# Development of an In-House EphA2 ELISA for Human Serum and Measurement of Circulating Levels of EphA2 in Hypertensive Patients with Renal Dysfunction

**DOI:** 10.3390/diagnostics12123023

**Published:** 2022-12-02

**Authors:** Maki Murakoshi, Takumi Iwasawa, Takeo Koshida, Yusuke Suzuki, Tomohito Gohda, Kazunori Kato

**Affiliations:** 1Department of Nephrology, Faculty of Medicine, Juntendo University, Tokyo 113-8421, Japan; 2Graduate School of Science and Engineering, Toyo University, Saitama 350-8585, Japan; 3Atopy Research Center, Graduate School of Medicine, Juntendo University, Tokyo 113-8421, Japan

**Keywords:** biomarker, hypertension, chronic kidney disease, eGFR, EphA2

## Abstract

Identifying novel biomarkers of kidney function in patients with chronic kidney disease (CKD) has strong clinical value as current measures have limitations. This study aims to develop and validate a sensitive and specific ephrin type-A receptor 2 (EphA2) enzyme-linked immunosorbent assay (ELISA) for human serum, and determine whether its results correlate with traditional renal measures in patients with hypertension. The novel ELISA of the current study was validated and used to measure circulating EphA2 levels in 80 hypertensive patients with and without kidney function decline (eGFR less than 60 mL/min/1.73 m^2^). Validation of the EphA2 ELISA showed good recovery (87%) and linearity (103%) and no cross-reactivity with other Eph receptors. Patients with kidney function decline had lower diastolic blood pressure, and higher UPCR and EphA2 than those without kidney function decline. The association of age and eGFR with EphA2 was maintained in the stepwise multiple regression analysis. In a multivariate logistic model, EphA2 was associated with a lower eGFR (<60 mL/min/1.73 m^2^) after adjustment for age, sex, and UPCR. High circulating EphA2 levels have potential application as a clinical biomarker for the presence of CKD in patients with hypertension.

## 1. Introduction

Elevated blood pressure is a leading cause of chronic kidney disease (CKD) and end-stage kidney disease (ESKD). CKD has been identified as one of the world’s most critical health issues with increasing global morbidity and mortality [1]. People with CKD experience an enhanced risk of cardiovascular disease, kidney failure, and mortality, despite interventions to manage risk factors. The common standardized assessments for risk of kidney disease progression are measurements of the estimated glomerular filtration rate (eGFR) and proteinuria/albuminuria. Substantial kidney damage, however, could be occurring in the early stages of CKD before noticeable decrements in eGFR. Therefore, more sensitive and specific CKD biomarkers other than albuminuria and eGFR are greatly needed.

We have previously demonstrated that high levels of TNF-related biomarkers such as TNF, progranulin (PGRN) [2], and TNF receptors (TNFRs) [3,4,5] predict future eGFR decline in both diabetic and non-diabetic CKD patients independent of traditional renal measures such as albuminuria and eGFR. High circulating levels of progranulin (PGRN) have been previously demonstrated to be negatively associated with eGFR [6] and future eGFR declines even after adjustment for baseline eGFR and proteinuria in patients with diabetes [2]. PGRN is a secreted glycoprotein that has been implicated in multiple pathological processes, e.g., the regulation of inflammation, embryonic development, tissue repair, tumorigenesis, and lysosomal function [7,8]. PGRN also binds to ephrin type-A receptor 2 (EphA2) [9], indicating that EphA2 levels may associate with traditional renal measures. EphA2 is a well-known biomarker for cancer diseases [10,11,12].

Eph receptors constitute the largest known family of receptor tyrosine kinases, comprising at least 16 distinct receptors [13]. Park et al. [14] reported that EphA2 has 55–70% and 40–50% sequence homologies and sequence identities, respectively, with other Eph receptors at the amino acid level. Based on this information, a more specific and sensitive assay is needed. Most commercial enzyme-linked immunosorbent assay (ELISA) kits for EphA2 have utilized polyclonal antibodies until now. Monoclonal antibodies are more specific to a single epitope and are less likely to bind nonspecifically to other antigens than polyclonal antibodies.

This study aims to develop and validate a human EphA2 ELISA using a sensitive monoclonal antibody and investigate the relationship between proteinuria, eGFR, and EphA2 in hypertensive patients.

## 2. Materials and Methods

### 2.1. Monoclonal Antibodies to Human EphA2

Hybridomas producing mAbs 230-1 and 57-1 against human EphA2 were generated by immunizing BALB/c mice (The Jackson Laboratory Japan, Yokohama, Japan) with human EphA2 proteins purified from supernatants of HEK239-F (Invitrogen, Thermo Fisher Scientific, Waltham, MA, USA) transfected with pcDNA3.1 encoding human EphA2. The immunized splenocytes were then fused with P3X63Ag8U mouse myeloma cells (JCRB Cell Bank, Osaka, Japan). The supernatants were harvested 14 days postfusion, and then screened in reactivity against EphA2 transfectants, but not against ErbB2 transfectants with a cell-based ELISA system. After cloning by limiting dilutions twice, hybridoma cell colonies termed 230-1 and 57-1 (mouse IgG1, κ) were selected. To confirm this, the reactivity of mAbs 230-1 and 57-1 to EphA2 transfectants was examined with flow cytometry (FACSCalibur, BD Immunocytometry Systems, Franklin Lakes, NJ, USA).

### 2.2. Novel ELISA for Detecting of Human EphA2

A novel ELISA was developed for detecting serum EphA2 levels. A 96-well ELISA plate (Cat. # 3590; Corning, NY, USA) was coated with 5.0 µg/mL of captured (human anti-EphA2) antibody (230-1). The plate was maintained overnight at 4 °C and then blocked with SuperBlock Blocking Buffer (Thermo Fisher Scientific, Waltham, MA, USA) for 24 h at 4 °C. A 100-mL serum sample or standard (recombinant human EphA2 protein, Cat. # 3035-A2; R&D Systems, Minneapolis, MN, USA) was diluted with Can Get Signal Solution 1 (Cat. # NKB-201; TOYOBO, Osaka, Japan) to a final dilution of 1:5 and was then added in duplicate to separate wells. The wells were washed thrice with washing buffer (PBS-Tween 20 (0.05%); Merk, Darmstadt, Germany) after the plate was incubated at room temperature for 2 h. Thereafter, 100 μL of diluted (to a final concentration of 0.5 μg/mL) biotinylated detection antibody 57-1 in Can Get Signal Solution 2 (Cat. # NKB-301; TOYOBO, Osaka, Japan) was added to each well, and the plate was incubated at room temperature for 1 h. Wells were then washed four times. Then, 100 μL/well of diluted (to a final dilution of 1:1000) streptavidin poly horse-radish peroxidase (HRP, Cat. # 554066; Thermo Fisher Scientific, Waltham, MA, USA) in ELISA Assay Diluent (Cat. # 421203; BioLegend, San Diego, CA, USA) was added and maintained for 45 min at room temperature. Wells were then washed five times. TMB 1-component microwell peroxidase substrate, SureBlue (Cat. # 5120-0075; SeraCare Life Sciences Inc., Milford, MA, USA) at 100 μL/well was added and maintained for 15 min at room temperature. Sulfuric acid was added at 100 μL/well to stop the reaction. The absorbance values of each well were measured using an automated microplate reader (SpectraMax, Molecular Devices, Sun Valley, CA, USA) with compatible software (SoftMax Pro, version 5.4.1, Molecular Devices, Sun Valley, CA, USA) with measurement and reference wavelengths of 450 nm and 570 nm, respectively. The optical density (OD) of the blank well was subtracted from each sample OD. The initial segment of the standard curve was observed to be linear with a second-degree polynomial.

### 2.3. Validation Procedures

#### 2.3.1. Specificity

The novel in-house EphA2 ELISA system of the current study was verified by applying four different concentrations (250, 500, 1000, and 2000 pg/mL) of recombinant EphA2 (Cat. # 3035-A2; R&D Systems, Minneapolis, MN, USA) to determine whether the assay worked properly. The specificity of the novel EphA2 ELISA assay was examined next by demonstrating the absence of an increased signal after the addition of the same concentrations (250, 500, 1000, and 2000 pg/mL) of recombinant human EphA1 (Cat. # 7146-A2; R&D Systems, Minneapolis, MN, USA), EphA3 (Cat. # 6444-A3; R&D Systems, Minneapolis, MN, USA), or EphA4 (Cat. # 6827-A4; R&D Systems, Minneapolis, MN, USA).

#### 2.3.2. Accuracy

Accuracy was estimated with a spike and recovery assay and dilution linearity evaluation. Serum samples (50 μL; *n* = 6) with a wide range of EphA2 levels [i.e., low (300–500 pg/mL), medium (500–1500 pg/mL), and high (1500–4000 pg/mL)] were spiked with 10 μL of known amounts of recombinant EphA2 [unspiked: 0 pg/mL, low: 1200 pg/mL (final concentration, 200 pg/mL), medium: 3600 pg/mL (final concentration, 600 pg/mL), and high 10,800 pg/mL (final concentration 1800 pg/mL). The 60 μL samples were diluted with Can Get Signal Solution 1 to a final dilution of 1:5 and the assay was performed. The percent recovery was calculated using the following formula: (observed spiked sample value − unspiked sample value)/(actual amount spiked in sample) × 100.

Sixteen serum samples with a wide range of EphA2 levels [low (150–500 pg/mL), medium (500–1000 pg/mL), high (1000–2000 pg/mL), and very high (2000–4000 pg/mL)] were used to evaluate the linearity of dilutions. The serum samples were serially (two-, four-, eight-, and 16-fold dilution) diluted with Can Get Signal Solution 2 to determine the optimal working dilutions of the serum EphA2 analysis. Percent recovery was calculated using the following formula: (observed value at dilution)/(expected value after dilution) × 100.

#### 2.3.3. Lower Limit of Detection and Quantitation Limit

The lower limit of detection and the lower quantitation limit was calculated using the formula: 3.3 × [standard deviation (SD) of blank OD/the gradient of a standard curve around the lower limit of the detection] and 10 × (SD of blank OD/the gradient of a standard curve around the lower limit of the detection), respectively.

#### 2.3.4. Precision

Precision was expressed as the percent coefficient of variation (CV) for both intra- and inter-assay variability. Intra-assay variation was calculated by running 16 samples (in duplicate) on a single plate. Moreover, the inter-assay variation was determined by running 16 samples across five different days.

#### 2.3.5. Measurement with a Commercially Available EphA2 ELISA

Serum EphA2 levels was also measured using a commercially available ELISA kit (Cat. # EH173RB; Thermo Fisher Scientific, Waltham, MA, USA) to compare the novel in-house EphA2 ELISA system of the current study with the standard system.

### 2.4. Patients and Sample Collection

Sixteen hypertensive patients with normal renal function (eGFR ≥ 60 mL/min/1.73 m^2^) and 64 patients with renal dysfunction (eGFR < 60 mL/min/1.73 m^2^) were enrolled in the present study. Individuals with diabetes and glomerulonephritis at baseline were excluded.

Serum samples were stored at −80 °C until the analysis was performed. The study protocol and informed written consent procedures were approved by the Institutional Review Board of the Juntendo University Faculty of Medicine, Tokyo, Japan (IRB No. M19–0223). Informed consent was obtained from all participants included in the study.

### 2.5. Patients and Sample Collection

Normally, distributed continuous variables are expressed as the mean ± SD and were compared using Student’s *t*-test. Skewed continuous variables were handled as continuous variables after common logarithmic transformation and then presented as the median (25th and 75th percentile). A Mann–Whitney *U* test was performed for comparisons of continuous variables. Correlations between two continuous variables were assessed using Spearman’s rank-order correlation test. Stepwise multiple linear regression was performed to determine the factor that affects the EphA2 level. Logistic regression analysis was performed to determine the factors that predict a lower eGFR (<60 mL/min/1.73 m^2^). A two-sided *p* value < 0.05 was considered statistically significant. All statistical analyses were performed with the SAS Enterprise Guide (version 8.1; SAS Institute, Cary, NC, USA).

## 3. Results

### 3.1. Validation of the Novel In-House EphA2 ELISA

Sufficient concentrations of recombinant EphA1, EphA2, EphA3, and EphA4 proteins were added to separate wells of a plate to exclude the possibility of artifacts due to cross-reactivity among EphA receptor antibodies. The in-house EphA2 ELISA system of the current study showed remarkable specificity for EphA2, and no increased signals for EphA1, EphA3, and EphA4 were noted (Appendix A).

The lower detection and lower quantitation limits were found to be 9.7 and 29.5 pg/mL, respectively. The hook effect was not observed, at least at 4000 pg/mL. Therefore, the ELISA system of the current study could reliably detect human EphA2 with a working range from 29.5 to 4000 pg/mL (Appendix A).

Recovery was evaluated by the ability to recover a known amount of recombinant EphA2 protein (Appendix A). The percent recovery ranged from 79% to 87%, 81% to 97%, and 80% to 111% for low (200 pg/mL), medium (600 pg/mL), and high (1800 pg/mL) levels of spiked recombinant EphA2 solution, respectively. This indicated that the average percent recovery of the spiked samples was acceptable (87%).

Dilutional linearity was evaluated by serial dilution (two- to 16-fold) of a wide range of serum concentrations (197–3752 pg/mL, Appendix A). The percent recovery was excellent (103%).

The intra- and inter-assay variabilities for serum samples were 2.8% and 3.5%, respectively, indicating that the assay had excellent precision (Appendix A).

### 3.2. Comparison with the EphA2 ELISA Kit from Thermo Fisher Scientific

A moderate correlation was observed between the novel in-house system and a Thermo Fisher Scientific ELISA (*r* = 0.41; Appendix A). The detectability values of the novel in-house ELISA of the current study and the Thermo Fisher Scientific ELISA were 100% and 83.8%, respectively (Appendix A). Overall, EphA2 concentrations measured with the in-house ELISA of the current study were higher than those of the Thermo Fisher Scientific ELISA. The correlation between eGFR and EphA2 measured with the in-house kit was strong compared to that measured with the Thermo Fisher Scientific ELISA (Figure 1).

### 3.3. Circulating EphA2 Levels in Patients with Hypertension

The characteristics of the study subjects are presented in Table 1. Their mean age was 72 years old and 52 of 80 (65.0%) were males. The difference between the mean age and systolic blood pressure of the two groups did not reach statistical significance. Patients with decreased eGFR had a significantly higher UPCR and EphA2 concentration, and lower diastolic blood pressure than those with a normal eGFR. A total of 54 (67.5%) patients were treated with renin-angiotensin system (RAS) blockers. As shown in Table 2, the serum level of EphA2 was correlated with age, DBP, eGFR, and UPCR. The association of age and eGFR with EphA2 was maintained in the stepwise multiple regression analysis (Table 3). Furthermore, EphA2 was a predominant determinant of eGFR < 60 mL/min/1.73 m^2^ after adjustment for age, sex, and UPCR in multivariate logistic regression analysis (model 1, Table 4). In sensitivity analysis, EphA2 remained significant even after additional adjustment for DBP or BP control (model 2 and 3, Table 4). Similar results were obtained when eGFR < 30 mL/min/1.73 m^2^ was used as an objective variable instead of eGFR < 60 mL/min/1.73 m^2^ (Table 5).

## 4. Discussion

In the current study, a novel in-house ELISA for EphA2 was developed and validated to assess serum EphA2 concentrations of hypertensive patients. To achieve this aim, the determination of lower detection and quantitation limits, specificity, dilution linearity, spike recovery, and precision experiments were included. Eph receptors are divided into EphA and EphB subfamilies based on their ligand specificity. EphA2 is localized on the cell membrane [15] and binds to ephrin-A1, -A2, -A3, -A4, and -A5 [16,17]. The novel assay showed no cross-reactivity with other EphA receptors, e.g., EphA1, EphA3, and EphA4. The novel ELISA of the current study passed the criteria for dilution linearity with ease. Spike recovery was about 79–111% for all concentrations and was within the accepted range of 80–120% with two exceptions where the limits were slightly exceeded (spike recovery for low spikes in medium or high levels of serum samples). Precision inter- and intra-assay CV values were excellent (<5%). The lower quantitation limit of 29.5 pg/mL was significantly lower than the amount of EphA2 observed in all serum samples of the study patients. The detectability of the in-house ELISA system of the current study was superior to the commercial EphA2 ELISA kit (Thermo Fisher Scientific). The EphA2 concentration determined with the Thermo Fisher Scientific ELISA kit were generally lower than those of the in-house ELISA. When the standard reagent accompanying the Thermo Fisher Scientific ELISA kit was adjusted to 2000 pg/mL with a dilution buffer and then measured with the in-house buffer;, the concentrations were almost comparable [average concentration: 2058 pg/mL (*n* = 2)]. The moderate (*r* = 0.41) correlation between the assay of the current study and the Thermo Fisher Scientific assay may indicate that the two ELISA methods detect different alterations in EphA2 serum levels.

Recent investigations revealed that EphA2 is a key modulator for a wide variety of cellular functions. EphA2 is also a cardiovascular molecule involved in the regulation of cell–cell interactions and angiogenesis [18]. EphA2 expression has been detected in a wide assortment of tissues including the kidney [19]. Satake et al. [20] reported that EphA2 was primarily expressed in proximal tubules, with minimal staining in glomeruli, distal tubules, interstitium, and vessels. Several studies reported an increase in EphA2 expression under stressful conditions, e.g., an injury- and stress-response regulation [21,22]. For example, Baldwin et al. [23] showed upregulation of EphA2 and ephrinA1 in mouse kidneys following renal ischemia-reperfusion injury. The current study first determined the features of circulating EphA2 levels in patients with hypertension and analyzed the association between renal function and EphA2 levels. A strong negative correlation was observed between EphA2 concentrations and eGFR. Hypertension is one of the leading causes of kidney function decline due to the deleterious effects that increased blood pressure has on the kidney vasculature. Long-term hypertension is considered to increase intraglomerular pressure and impair glomerular filtration. Furthermore, novel evidence demonstrated that persistent high blood pressure injures tubular cells, leading to epithelial and tubulointerstitial fibrosis [24]. Considering that EphA2 is expressed in the proximal tubules, elevated EphA2 concentrations in patients with a decline in kidney function may chiefly reflect tubular damage. In addition to those mentioned above, EphA2 is upregulated at the gene and protein levels in a variety of human tissues specimens and cancer cell lines. Therefore, we should pay attention to the interpretation of results for measurement of EphA2 in patients with comorbid cancer [25].

## 5. Conclusions

A novel assay that reliably detects EphA2 in human sera was developed. The assay has several strengths including high sensitivity and specificity. Circulating EphA2 levels were significantly higher in patients with renal function decline compared with patients with normal kidney function, suggesting the potential diagnostic efficiency of circulating EphA2 levels in evaluating kidney function in patients with hypertension. Further studies are needed to find out if EphA2 levels are associated with renal measures in patients with other types of kidney disease and whether they can predict future renal function decline. Additionally, it is necessary to verify whether EphA2 is useful in predicting renal function decline compared to previously reported candidate CKD biomarkers [2,3,4,5]. Revealing the roles of EphA2 and the other types of Eph receptors in kidney disease will be important in the future.

## Figures and Tables

**Figure 1 diagnostics-12-03023-f001:**
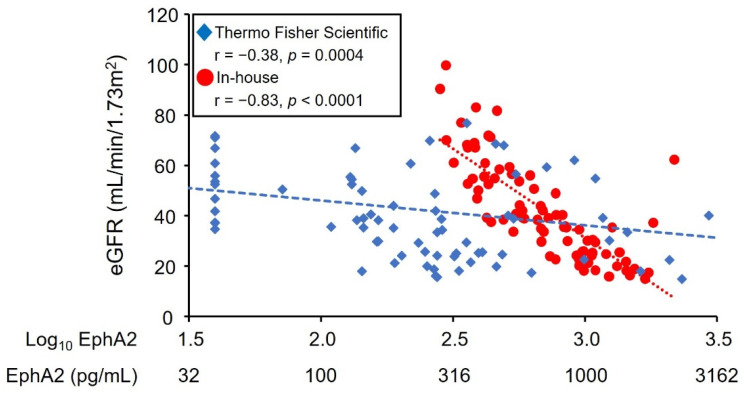
The correlation between EphA2 concentrations and eGFR. Spearman’s correlation coefficients revealed that EphA2 levels had a significant negative correlation with eGFR.

**Table 1 diagnostics-12-03023-t001:** Clinical characteristics and serum EphA2 levels in patients with hypertension.

Characteristics	Total (*n* = 80)	eGFR (mL/min/1.73 m^2^)		*p*
		<60 (*n* = 64)	≥60 (*n* = 16)	(<60 vs. ≥60)
eGFR, mL/min/1.73 m^2^	39 (26, 56)	35 (24, 43)	71 (67, 79)	By design
eGFR categories, no				
G1+2 (≥60 mL/min/1.73 m^2^)	16			
G3a (45–59 mL/min/1.73 m^2^)	14			
G3b (30–44 mL/min/1.73 m^2^)	26			
G4 (15–29 mL/min/1.73 m^2^)	24			
Age, year	72 ± 11	73 ± 11	68 ± 6	0.13
Male, no (%)	52 (65.0)	43 (67.2)	9 (56.3)	0.42
SBP, mmHg	128 ± 13	128 ± 13	125 ± 11	0.42
DBP, mmHg	73 ± 10	72 ± 10	78 ± 8	0.03
RAS blockers Tx, no (%)	54 (67.5)	43 (67.2)	11 (68.8)	0.91
UPCR, g/gCr	0.38 (0.10, 1.00)	0.50 (0.19, 1.46)	0.09 (0.06, 0.26)	<0.0001
EphA2, pg/mL	682 (434, 1006)	775 (564, 1061)	382 (329, 430)	<0.0001

Data are presented as the mean ± SD, median (25th and 75th percentile), or percentage. eGFR, estimated glomerular filtration rate; SBP, systolic blood pressure; DBP, diastolic blood pressure; RAS, renin-angiotensin system; Tx, therapy; UPCR, ratio of proteinuria to creatinine; EphA2, Eph-receptor tyrosine kinase-type A2.

**Table 2 diagnostics-12-03023-t002:** Spearman’s rank-order correlation between serum EphA2 concentrations and various clinical parameters in patients.

Vendor	In-House		Thermo Fisher Scientific	
EphA2	*r*	*p*	*r*	*p*
Age	0.37	0.0007	0.03	0.82
SBP	0.16	0.16	0.19	0.08
DBP	−0.26	0.02	0.07	0.57
eGFR	−0.83	<0.0001	−0.38	0.0004
UPCR	0.61	<0.0001	0.23	0.04

Abbreviations used in this table are the same as in Table 1.

**Table 3 diagnostics-12-03023-t003:** Stepwise multiple regression analysis of factors associated with EphA2.

Variable	*β*	*F*	*p*
Age	0.003	4.69	0.03
eGFR	−0.156	76.91	<0.0001
UPCR	0.031	3.33	0.07
DBP	0.0003	0.03	0.86
*R^2^*	0.70		

*F* value > 4.0 was considered significant. Abbreviations used in this table are the same as in Table 1.

**Table 4 diagnostics-12-03023-t004:** Multivariate logistic regression model of risk for lower eGFR in hypertensive patients according to clinical predictors and serum EphA2.

Baseline Characteristics (Units of Increase)	Model 1		Model 2		Model 3	
	OR (95% CI)	*p*	OR (95% CI)	*p*	OR (95% CI)	*p*
Sex (female)	0.96 (0.21, 4.50)	0.96	1.02 (0.21, 4.92)	0.98	0.82 (0.15, 4.50)	0.82
Age (10 year)	1.33 (0.50, 3.57)	0.57	1.21 (0.40, 3.66)	0.74	1.35 (0.50, 3.63)	0.55
UPCR (1 quartile)	3.79 (1.31, 10.99)	0.014	3.98 (1.32, 12.02)	0.01	3.81 (1.32, 11.00)	0.01
EphA2 (1 quartile)	4.74 (1.46, 15.42)	0.0096	4.48 (1.36, 14.72)	0.01	4.84 (1.47, 15.93)	0.01
DBP (10 mmHg)			0.74 (0.36, 1.53)	0.42		
BP control (poor)					1.58 (0.23, 11.06)	0.64

Abbreviations used in this table are the same as in Table 1. OR, Odds ratio. Poor BP control was defined as BP of ≥140/90 mmHg.

**Table 5 diagnostics-12-03023-t005:** Multivariate logistic regression model of risk for lower eGFR <30 mL/min/1.73 m^2^ in hypertensive patients according to clinical predictors and serum EphA2.

Baseline Characteristics (Units of Increase)	Model 1		Model 2		Model 3	
	OR (95% CI)	*p*	OR (95% CI)	*p*	OR (95% CI)	*p*
Sex (female)	0.45 (0.08, 2.43)	0.35	0.39 (0.07, 2.17)	0.28	0.33 (0.06, 1.95)	0.22
Age (10 year)	0.74 (0.40, 1.38)	0.45	0.68 (0.36, 1.30)	0.24	0.88 (0.46, 1.66)	0.68
UPCR (1 quartile)	2.43 (1.09, 5.43)	0.03	2.45 (1.07, 5.60)	0.03	2.79 (1.16, 6.75)	0.02
EphA2 (1 quartile)	8.78 (2.68, 28.78)	0.0003	10.42 (2.71, 40.07)	0.0006	9.56 (2.81, 32.48)	0.0003
DBP (10 mmHg)			0.59 (0.24, 1.47)	0.26		
BP control (poor)					0.25 (0.03, 1.90)	0.18

Abbreviations used in this table are the same as in Table 1. OR, Odds ratio. Poor BP control was defined as BP of ≥140/90 mmHg.

## Data Availability

Not applicable.

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
