# Peer review of "Development of an In-House EphA2 ELISA for Human Serum and Measurement of Circulating Levels of EphA2 in Hypertensive Patients with Renal Dysfunction"

_diagnostics, 2022, doi:10.3390/diagnostics12123023_

Round 1

Reviewer 1 Report

I find the presented for review new assay of  in-house ELISA test for EphA2 useful, and I have no objectives against the test vvalidation. However, I think that the paper should contain also the potential limitations of test use. EphA2 is expressed in many cells, and as such the obtained results may be not necessery true for chronic kidney disease. This should be stated in discussion or study limitations when EphA2 test should not be performed, e.g. in cancer diseases.

Author Response

As suggested by the reviewer, EphA2 is upregulated at the gene and protein levels in a variety of human tumor tissues specimens and cancer cell lines. Therefore, we should pay attention to the interpretation of results for measurement of EphA2 in patients with comorbid cancer. We have added these limitations to the revised the Discussion section of our manuscript (pages 8, lines 279-282).

Reviewer 2 Report

Murakoshi et al submit an original research article entitled "Development of an in-house EphA2 ELISA for human serum and measurement of circulating levels of EphA2 in hypertensive patients with renal dysfunction". They studied a cohort of patients with renal function decline compared with patients with normal kidney function, to establish the potential diagnostic efficiency of a novel assay that reliably detects EphA2 in human sera . The assay had several strengths including high sensitivity and specificity. Circulating EphA2 levels were significantly higher in patients with renal function decline compared with patients with normal kidney function, suggesting the potential diagnostic efficiency.

The publication is well written and the material and method section is comprehensive. One question is "Sixteen hypertensive patients with normal renal function and 64 patients with renal dysfunctionwere enrolled in the  present study. " Why is there such a disproportion between the 2 groups?

Other question "Individuals with diabetes and glomerulonephritis at baseline were excluded." What is the reason of these exclusions?

The discussion should better take into account other alternative as CKD biomarkers, compared to eFGR and EphA2 (N-Gal etc.).

A recapitulative figure would be welcome.

Author Response

  1. To examine the association with renal function, we included the patients with various kinds of GFR categories (G1+2/G3a/G3b/G4, n=16/14/26/24, respectively). This resulted in a disproportion in the number of patients when divided by greater or less than eGFR 60 mL/min/1.73 m2. The multivariate logistic regression analysis also analyzed the results when divided by greater or less than eGFR 30 mL/min/1.73 m2 and confirmed a similar trend. We have added the CKD classification data to Table 1.
  2. In order to evaluate in patients with same condition in terms of pathophysiological classification, patients with diabetes or glomerulonephritis were excluded in the present study.

  3. It is indeed important to investigate the association between EphA2 and other biomarkers of kidney function. Unfortunately, the current study did not investigate renal biomarkers other than GFR and proteinuria. We have added these limitations to the Conclusions (pages 8, lines 291-292).

  4. We have submitted a recapitulative figure.